# Phosphatidylserine-Exposing Annexin A1-Positive Extracellular Vesicles: Potential Cancer Biomarkers

**DOI:** 10.3390/vaccines11030639

**Published:** 2023-03-13

**Authors:** Gloria I. Perez, Matthew P. Bernard, Daniel Vocelle, Ahmed A. Zarea, Najla A. Saleh, Matthew A. Gagea, Doug Schneider, Maxine Bauzon, Terry Hermiston, Masamitsu Kanada

**Affiliations:** 1Institute for Quantitative Health Science and Engineering (IQ), Michigan State University, East Lansing, MI 48824, USAmbernard@msu.edu (M.P.B.);; 2College of Osteopathic Medicine, Michigan State University, East Lansing, MI 48824, USA; 3Department of Pharmacology & Toxicology, Michigan State University, East Lansing, MI 48824, USA; 4Cell and Molecular Biology Program, Michigan State University, East Lansing, MI 48824, USA; 5College of Natural Science, Michigan State University, East Lansing, MI 48824, USA; 6Lyman Briggs College, Michigan State University, East Lansing, MI 48824, USA; 7GLAdiator Biosciences, Mill Valley, CA 94941, USA; 8College of Human Medicine, Michigan State University, East Lansing, MI 48824, USA

**Keywords:** extracellular vesicle, phosphatidylserine, cancer, biomarker, annexin A1, CD63, Gla domain, annexin A5, GlaS, protein S, flow cytometry

## Abstract

Under physiological conditions, phosphatidylserine (PS) predominantly localizes to the cytosolic leaflet of the plasma membrane of cells. During apoptosis, PS is exposed on the cell surface and serves as an “eat-me” signal for macrophages to prevent releasing self-immunogenic cellular components from dying cells which could potentially lead to autoimmunity. However, increasing evidence indicates that viable cells can also expose PS on their surface. Interestingly, tumor cell-derived extracellular vesicles (EVs) externalize PS. Recent studies have proposed PS-exposing EVs as a potential biomarker for the early detection of cancer and other diseases. However, there are confounding results regarding subtypes of PS-positive EVs, and knowledge of PS exposure on the EV surface requires further elucidation. In this study, we enriched small EVs (sEVs) and medium/large EVs (m/lEVs) from conditioned media of breast cancer cells (MDA-MB-231, MDA-MB-468) and non-cancerous cells (keratinocytes, fibroblasts). Since several PS-binding molecules are available to date, we compared recombinant proteins of annexin A5 and the carboxylated glutamic acid domain of Protein S (GlaS), also specific for PS, to detect PS-exposing EVs. Firstly, PS externalization in each EV fraction was analyzed using a bead-based EV assay, which combines EV capture using microbeads and analysis of PS-exposing EVs by flow cytometry. The bulk EV assay showed higher PS externalization in m/lEVs derived from MDA-MB-468 cells but not from MDA-MB-231 cells, while higher binding of GlaS was also observed in m/lEVs from fibroblasts. Second, using single EV flow cytometry, PS externalization was also analyzed on individual sEVs and m/lEVs. Significantly higher PS externalization was detected in m/lEVs (annexin A1^+^) derived from cancer cells compared to m/lEVs (annexin A1^+^) from non-cancerous cells. These results emphasize the significance of PS-exposing m/lEVs (annexin A1^+^) as an undervalued EV subtype for early cancer detection and provide a better understanding of PS externalization in disease-associated EV subtypes.

## 1. Introduction

Phosphatidylserine (PS) is a negatively charged phospholipid found on the inner leaflet of the cell membrane under physiological conditions. Upon initiating the apoptotic cascade, enzymes such as scramblases can disrupt the asymmetric PS distribution, resulting in the externalization of PS. This externalized PS has been proposed as an “eat-me” signal for PS receptors on macrophages and promotes clearance of apoptotic debris [1]. Furthermore, masking PS on apoptotic cells has been demonstrated to inhibit their engulfment by macrophages in vivo [2]. Interestingly, recent studies have shown that PS can be externalized on the surfaces of viable cells as a mechanism for eliciting immune evasion [3,4,5,6]. Previous studies have demonstrated that activated platelets, monocytes, mature macrophages, activated B cells, activated T cells, dendritic cells, infected cells (both viral and bacterial), tumor vasculature, tumor cells, and tumor cell-derived exosomes may externalize PS on their surfaces [3]. Moreover, constitutive PS exposure on viable cells did not induce phagocytosis, suggesting that macrophages could distinguish PS on viable and apoptotic cells [7]. However, the exact mechanism of this phenomenon remains largely unknown. Macrophage activation via oxidative oligomerization of protein S on the apoptotic cell surface has been proposed as a possible mechanism [8]. In addition, a recent study has shown that several chemokines bind PS on apoptotic cells and possibly attract phagocytes [9]. This finding suggests that PS exposure during apoptosis accompanies the binding of specific signaling molecules for anti-inflammatory responses.

Extracellular vesicles (EVs) are released from cells into body fluids [10,11]. EVs are heterogeneous in biogenesis, size, and cargo [12,13]. Exosomes (40–120 nm in diameter) are a major class of EVs produced in endosomal compartments called multivesicular bodies. Another EV class, microvesicles or ectosomes (50–1000 nm in diameter), are formed by the outward budding of the plasma membrane. EVs from tumor cells have recently received considerable attention for their potential usage as a liquid biopsy since they carry biomarkers such as cancer-specific proteins and nucleic acids indicative of cancer development [14,15]. Interestingly, cells in culture abundantly expose PS on the surface of EVs. Moreover, quantitative assessment of PS-positive exosomes derived from cancer cells allowed for the detection of early-stage malignancies before any other clinical evidence of the disease appeared [16,17]. Similarly, PS-positive EVs in plasma, released mainly from platelets and endothelial cells, alternatively called microparticles, have been studied as potential biomarkers for diagnosing various diseases, including cancer [18], cardiovascular diseases [19], liver damage [20], and COVID-19 disease severity [21]. However, inherent heterogeneities of EVs, such as sizes, membrane compositions, and cargo molecules, pose significant challenges to establishing reliable biomarkers.

We have recently reported that two different PS-binding proteins, annexin A5 and a carboxylated glutamic acid domain of Protein S (GlaS) [22], bound EVs isolated from murine tumor cells [23]. In the current study, we further characterized the binding capacity of annexin A5 and GlaS to different EV classes, small EVs (sEVs; <200 nm in diameter) and medium/large EVs (m/lEVs; >200 nm in diameter), categorized according to the Minimal Information for Studies of Extracellular Vesicles (MISEV) 2018 guidelines [24], derived from cancer and non-cancerous cell types. Currently, most studies focus on sEVs as potential biomarkers, while m/lEVs are considered undervalued subtypes. Therefore, we first performed bead-based bulk EV assays utilizing flow cytometry to investigate the average properties of PS externalization on the surface of sEVs and m/lEVs using annexin A5 and GlaS proteins. Despite the unique challenges in analyzing submicron-sized EVs compared to much larger cells, recent studies have demonstrated reliable single EV flow cytometric analyses [25]. However, to date, the applicability of this approach to different EV classes (e.g., sEVs vs. m/lEVs) has not been rigorously characterized. Thus, we further aimed to compare the binding efficiencies of annexin A5 and GlaS proteins to PS-exposing sEVs and m/lEVs, identified by CD63 or annexin A1 positivity, respectively, derived from cancer and non-cancerous cells using single EV flow cytometry. Here, we anticipate these studies demonstrate the significance of m/lEVs, in addition to sEVs, as reliable biomarkers for analyzing individual PS-exposing EVs by flow cytometry due to their larger surface area.

## 2. Materials and Methods

### 2.1. Cell Culture

MDA-MB-231 cells (ATCC) and MDA-MB-468 cells (ATCC) were cultured in Dulbecco’s Modified Eagle Medium (DMEM) supplemented with 10% (vol/vol) FBS and 1% penicillin/streptomycin. Human Epidermal Keratinocytes (Pooled; A13401; Gibco, New York, NY, USA) were cultured in EpiLife Medium (MEPI500CA, Gibco) supplemented with Human Keratinocyte Growth Supplement (S0015, Gibco, New York, NY, USA) and 1% penicillin/streptomycin. BJ fibroblasts (ATCC) were cultured in Eagle’s Minimum Essential Medium (EMEM) with 10% FBS and 1% penicillin/streptomycin. Cells were incubated at 37 °C in a 5% CO_2_ atmosphere.

### 2.2. EV Isolation

EV-depleted FBS was prepared by ultracentrifugation at 100,000× *g*, 4 °C for 18 h [26]. MDA-MB-231 cells, MDA-MB-468 cells, keratinocytes, and fibroblasts were seeded at 1.5 × 10^6^ cells per 100-mm dish. On the following day, the media were changed to EV-depleted media (except for keratinocytes because of the serum-free medium), and the cells were cultured for an additional 72 h. The conditioned media were collected, and sEV- and m/lEV-enriched fractions were isolated as previously described [27,28]. Briefly, the conditioned media were centrifuged at 600× *g* for 5 min to remove cells. Next, the supernatants were centrifuged at 2000× *g* for 20 min to remove apoptotic bodies. Subsequently, m/lEV fractions were isolated by centrifugation at 20,000× *g* for 60 min at 4 °C and washed in PBS, then centrifuged at 20,000× *g* for 60 min at 4 °C and re-suspended in PBS. The supernatants were further filtered through 0.2 µm PES membrane filters (Nalgene, Rochester, NY, USA, 725-2520) to remove large vesicles. Finally, sEV fractions were collected by a size-based EV isolation method using 50-nm porous membranes (Whatman, Maidstone, UK, WHA110603) with holders (EMD Millipore, Billerica, MA, USA, SX0002500) by applying a vacuum pressure. The concentrated sEV fractions on the membranes were washed with 5 mL PBS and carefully collected from the membranes. The EV-enriched fractions’ protein concentrations and particle numbers were analyzed by Bradford assay and nanoparticle tracking analysis (NTA; ZetaView, Particle Metrix, Meerbusch, Germany), respectively. All EV fractions were aliquoted and stored at −80 °C.

### 2.3. Nanoparticle Tracking Analysis

The isolated sEV- and m/lEV-fractions were analyzed for particle size by NTA, following the manufacturer’s instructions (camera sensitivity = 85 or 89, shutter = 250 or 400, and frame rate = 30 or 7.5 for the analysis of sEV- or m/lEV-fractions, respectively; cutoffs = 10-nm minimum and 1000-nm maximum, minimum brightness = 22). EVs were diluted 100–1000 times with PBS to obtain ideal EV concentrations for measurements (1.0 × 10^9^–1.0 × 10^10^ EVs/mL).

### 2.4. Recombinant GlaS Protein Production and Fluorescent labeling

We have previously developed and characterized the fluorescently labeled GlaS proteins [23]. Plasmid pEAKflCMV (Eagle Scientific, Sandiacre, UK) expressing 283 amino acids from the N-terminus of human Protein S (signal peptide, propeptide, Gla domain, and 4 EGF-like domains) with an additional 6 × His-tag was used to express GlaS proteins in CHO cells. The recombinant GlaS proteins were purified using Ni-NTA columns. The GlaS proteins were analyzed by SDS-PAGE and HPLC. The purified GlaS proteins were labeled using a FITC protein labeling kit (F6434, Thermo Fisher, Waltham, MA, USA) and their protein concentrations were determined by Bradford assay.

### 2.5. Fluorescence Microscopy

Fluorescence deconvolved images were taken using the DeltaVision Microscope system (GE Healthcare Life Sciences, Piscataway, NJ, USA). EVs were observed using glass bottom chamber slides (80807, Ibidi). Isolated EVs were incubated with 0.8 µg/mL APC-labeled annexin A5 (640919, BioLegend, San Diego, CA, USA) and 4.8 µg/mL FITC-GlaS for 30 min at room temperature and directly observed by fluorescence microscopy. The fluorescence filter set FITC/Cy5 and the 60x oil immersion objective, 1.42 NA, were used to acquire images and process z-stacks (Optical section space: 0.2 µm, Number of optical sections: 30) for deconvolution. Maximum intensity projection images of z-stack were created using ImageJ software (Version 1.53a, NIH).

### 2.6. Western Blotting

The isolated sEVs and m/lEVs from 1.5 × 10^6^ cells following 72 h of cell culture were lysed with 4X sample buffer (Bio-Rad, Hercules, CA, USA) with β-mercaptoethanol (for detecting TSG101, flotillin−1, and annexin A1) or without β-mercaptoethanol (for detecting CD81, CD63, and CD9), ran on a 4–20% Mini-PROTEAN TGX gel (Bio-Rad) and transferred to PVDF membranes (IPFL00010, Millipore). The parental cells were also lysed and assessed as equal cell numbers. Membranes were blocked with PBS containing 5% skim milk and 0.05% Tween20 (*v/v*) for 30 min at room temperature; incubated with primary antibodies overnight at 4 °C at dilutions recommended by the suppliers as follows: anti-TSG101 (14497, Proteintech, Rosemont, IL, USA, 1:1000), anti-CD81 (66866, Proteintech, 1:3000), anti-CD9 (60232, Proteintech, 1:1000), anti-CD63 (10628D, Ts63, Thermo Fisher, 1:1000), anti-flotillin-1 (610820, BD Biosciences, 1:1000), and anti-annexin A1 (MAB37701, R&D Systems, Minneapolis, USA, 1:3000). Membranes were washed 3 times with PBS containing 0.05% Tween20 (*v/v*), incubated with HRP-conjugated anti-mouse (7076, Cell Signaling, Danvers, MA, USA, 1:10,000) or anti-rabbit (7074, Cell Signaling, 1:10,000) antibodies for 1 h at room temperature, and washed again to remove unbound antibodies. Membranes were visualized with ECL Select Western Blotting Detection Reagent (RPN2235, GE Healthcare) on ChemiDoc MP Imaging System (Bio-Rad).

### 2.7. Bead-Based Flow Cytometric Analysis of PS-Exposing sEVs and m/lEVs

The samples were prepared as previously reported [16] with slight modifications. The fractions of sEVs or m/lEVs (10 µg EV protein [16]) in 500 µL HEPES (25 mM)/PBS were mixed overnight at 4 °C with 5 µL of 4 µm aldehyde-activated latex beads (A37304, Invitrogen, Waltham, MA, USA). The beads were blocked with 200 µL of 1% BSA for 1 h at room temperature, followed by washing with HEPES/PBS at 5000× *g* for 5 min. The beads were resuspended in 100 µL of annexin binding buffer [10 mM HEPES (pH 7.4), 140 mM NaCl, 2.5 mM CaCl_2_] containing 0.8 µg/mL APC-labeled annexin A5 (640919, BioLegend) or 4.8 µg/mL FITC-labeled GlaS and mixed for 45 min at room temperature. Finally, the samples were diluted with 400 µL of 2% paraformaldehyde (PFA) and incubated for 15 min at room temperature before flow cytometry.

### 2.8. Single EV Flow Cytometric Analysis of Surface PS in sEVs and m/lEVs

Aliquots (125 µL) of sEV- or m/lEV-fractions in PBS were stained with 5 µM CellTrace Violet (CTV; C34571, Thermo Fisher) for 10 min at room temperature. Unreacted CTV dyes were removed with spin desalting columns (6564, BioVision, Waltham, MA, USA). Next, 10 µL of CTV-stained sEV fractions were mixed with 0.2% BSA, APC-labeled anti-CD63 antibodies (A15712, Invitrogen), 0.8 µg/mL FITC-annexin A5 (640905, BioLegend) or 4.8 µg/mL FITC-GlaS in annexin binding buffer [10 mM HEPES (pH 7.4), 140 mM NaCl, 2.5 mM CaCl_2_], while m/lEV fractions were mixed with 0.2% BSA, APC-labeled anti-annexin A1 antibodies (831603, BioLegend), 0.8 µg/mL FITC-annexin A5 (640905, BioLegend) or 4.8 µg/mL FITC-GlaS for 60 min at room temperature. PBS and PBS with CTV + FITC-annexin A5 or FITC-GlaS prepared without EVs (125 µL) were processed in the same manner and used as controls. Finally, the samples (20 µL) were diluted with 480 µL of 2% PFA and incubated for 15 min at room temperature before flow cytometry.

### 2.9. Flow Cytometer Settings and Data Analysis

EV-coated beads or CTV-labeled EVs were analyzed on a Cytek Aurora spectral cytometer within the Michigan State University Flow Cytometry Core Facility. Prior to each acquisition of CTV-labeled EVs, a Long Clean cycle was performed on the instrument, and Milli-Q water was run for a minimum of 30 min on high. CTV-labeled EVs were detected by utilizing a violet laser (405 nm) threshold (V3 detector; 451–466 nm) to coincide with CTV excitation/emission (405/450 nm). Threshold settings were determined based on PBS and PBS + CTV + FITC-annexin A5 staining reagents without EV controls. CTV-labeled EV samples were acquired for 10 s each to normalize acquisition time for all samples and controls. Signals from FITC and APC were unmixed using the SpectroFlo software (v3.0.1; Cytek Biosciences, California, CA, USA) using autofluorescence extraction and analyzed using FCS Express (v7; De Novo Software). Due to fluorescence thresholding settings, single beads were gated for the bead-based assays, while all events were analyzed in the CTV-labeled EV assays. Fluorescence minus 1 (FMO) controls were used as gating controls for both EV-coated bead and CTV-labeled EV assays with FITC (annexin A5 or GlaS) and APC (anti-CD63 or anti-annexin A1 antibodies) labeling.

### 2.10. Statistical Analysis

All statistical analyses were performed with GraphPad Prism 9 (GraphPad Software version 9). A 1-way ANOVA followed by Tukey’s post hoc test was used for the comparison of 3 or more groups. Error bars for all the graphs represent mean ± SD. *p* values < 0.05 were defined as significant.

## 3. Results

### 3.1. Isolation and Characterization of sEVs and m/lEVs Derived from Cancer and Non-Cancerous Cells

Whereas some studies have previously reported that cancer cells produce more EVs than non-cancerous cells, conflicting reports suggest caution in interpreting these findings, thus presenting no consensus to date [29]. To advance the knowledge of EV production in cancer and non-cancerous cells, we enriched sEVs and m/lEVs from conditioned media of two human breast cancer cell lines (MDA-MB-231 cells and MDA-MB-468 cells) and two human non-cancerous cell types (keratinocytes and fibroblasts; Figure 1A). Immortalized human cell lines, such as MCF10A cells [30], are often compared with breast cancer cells. However, studies have shown that these cells do not fully represent normal cells in the body [31]. Hence, in this study, we chose two primary human cell types, rather than immortalized cells, to better represent normal cells. The isolated EV fractions were characterized by nanoparticle tracking analysis (NTA) and EV protein quantification. As we previously reported [28,32], the size distributions of sEV- and m/lEV-enriched fractions overlap significantly (Figure 1B,C), indicating the difficulty of separating these EV subtypes based on the size and density of EVs. We performed the following studies under the constraints of our EV subtype separation methods. Importantly, we observed that the concentrations of cancer cell-derived sEVs were significantly higher than sEVs from non-cancerous cells, while the size distribution of sEVs derived from cancer and non-cancerous cells was similar. (Figure 1B). The concentration of sEVs from MDA-MB-231 cells were 1.7- and 10.8-fold higher than those of sEVs from keratinocytes and fibroblasts, respectively, whereas the concentration of sEVs from MDA-MB-468 cells was 3.8- and 23.7-fold higher than those of keratinocytes and fibroblasts, respectively. In contrast, these four cell types showed more heterogeneous size distributions in m/lEV-enriched fractions with distinct peaks and mean sizes (Figure 1C). The concentration of m/lEVs from MDA-MB-231 cells was 3.4- and 5.4-fold higher than keratinocytes and fibroblasts, respectively. The concentration of m/lEVs from MDA-MB-468 cells was 10.8- and 17.3-fold higher than keratinocytes and fibroblasts, respectively. The protein concentration of MDA-MB-468 cell-derived sEVs was exceptionally high. However, other EV fractions exhibited no significant differences between cancer and non-cancerous cells, and there was no correlation between EV protein concentrations and EV numbers (Figure 1B,C). Further, the enrichment of EV marker proteins, CD81, CD63, CD9, TSG101, flotillin-1, and annexin A1, was assessed in sEVs and m/lEVs released from each cell type (Figure 1D and Appendix A). The expression of these marker proteins in equal numbers of parental cells was also evaluated (Appendix A). As expected, sEV- or m/lEV-fractions derived from cancer cells showed higher enrichment of all the marker proteins than non-cancerous cell-derived sEVs. Notably, while tetraspanins were often used as markers for sEVs, significant signals in CD63 and CD9 were detected in cancer cell-derived m/lEVs, suggesting that these molecules are not unique markers for sEVs, as the MISEV 2018 guidelines stated [24]. Interestingly, CD81 and TSG101 were mainly detected in sEVs but not m/lEVs, suggesting they could be more specific for sEVs. In contrast, significantly higher signals in flotillin-1 and annexin A1 were detected in m/lEVs derived from cancer cells assessed relative to sEVs (Figure 1D). However, we cannot rule out that the sensitivity of the antibodies is below the detection limit. Together, these results suggested that cancer cells produce higher numbers of both sEVs and m/lEVs relative to non-cancerous cells.

### 3.2. Recognition of sEVs and m/lEVs by Fluorescently Labeled PS-Binding Annexin A5 and GlaS Proteins

Protein S consists of a prototypical N-terminal Gla domain, a thrombin-sensitive region (TSR), four epidermal growth factor (EGF)-like domains, and a sex hormone binding globulin (SHBG)-like domain at the C-terminus [22]. In this study, we designed the GlaS protein by deleting the SHBG domain to avoid promiscuous binding to other proteins and adding a His-tag to facilitate purification and detection (Figure 2A). Western blot analysis confirmed a single band under a non-reducing condition. In contrast, two major and two minor bands were detected under a reducing condition (Figure 2B). The lower band is the GlaS that has been cleaved at the TSR and runs separately due to disulfide reduction. The recombinant GlaS protein was further analyzed by size exclusion chromatography (HPLC-SEC) and showed a single peak (Figure 2C).

We have previously demonstrated that fluorescently labeled GlaS and annexin A5 proteins can bind EVs derived from tumor cells [23]. Here, we further investigated the binding of FITC-labeled GlaS and APC-labeled annexin A5 proteins to sEVs and m/lEVs derived from cancer and non-cancerous cells by fluorescence microscopy (Appendix A). In the droplets containing enriched sEVs derived from all four cell types, the punctate fluorescence was only visible around the edge of the droplets, possibly due to their weak fluorescence (Appendix A). In contrast, FITC and APC double-positive fluorescent puncta were readily observed in the droplets containing enriched m/lEVs from all the cell types assessed (Appendix A). Also, the overall fluorescence of APC-annexin A5 was relatively high in fibroblast-derived sEVs and m/lEVs, possibly reflecting fewer EVs enriched in these droplets (Appendix A). Notably, in addition to m/lEVs co-labeled with FITC-GlaS and APC-annexin A5, puncta of either FITC-GlaS or APC-annexin A5 were observed by fluorescence microscopy. These data suggested that FITC-GlaS and APC-annexin A5 might bind distinct subpopulations of PS-exposing m/lEVs. However, conventional fluorescence microscopy is suboptimal in assessing weak signals in individual sEVs and m/lEVs to investigate this possibility further.

### 3.3. Bead-Based Flow Cytometric Analysis of PS-Exposing sEVs and m/lEVs Using FITC-GlaS or APC-Annexin A5 Proteins

Characterizing individual EVs is challenging due to their small size [10]. However, bead-based EV analysis offers an inexpensive and simplified bulk method of semi-quantitative EV analysis that can be implemented in laboratories equipped with standard flow cytometers [33]. In the current study, a bead-based EV assay assessed the distinct recognition of PS-exposing EV subtypes using FITC-GlaS and APC-annexin A5 proteins. The fractions of sEVs and m/IEVs from cancer and non-cancerous cells were captured on aldehyde-activated latex beads, incubated with fluorescently labeled PS-binding proteins, and analyzed by flow cytometry (Figure 3A). In contrast to the characterization by fluorescence microscopy, PS-exposing sEVs captured on beads were readily recognized by both FITC-GlaS and APC-annexin A5 proteins (Figure 3B,C and Appendix A), suggesting the superior sensitivity of this approach for analyzing EVs.

FITC-GlaS showed no preferential binding to sEVs derived from these four cell types (Figure 3B and Appendix A). Unexpectedly, among the sEV-enriched fractions derived from the four cell types assessed, keratinocyte-derived sEVs, but not cancer cell-derived sEVs, showed significantly higher binding of APC-annexin A5 (Figure 3C and Appendix A). In contrast, FITC-GlaS showed higher binding to m/lEVs derived from MDA-MB-468 cells (94.6% positive) relative to m/lEVs from keratinocytes (61.8%), while it also showed higher binding to fibroblast-derived m/lEVs (94.2%; Figure 3D and Appendix A). Furthermore, APC-annexin A5 showed significantly higher binding to PS-exposing m/lEVs released from MDA-MB-468 cells (89.6% positive on beads) relative to m/lEVs from keratinocytes (51.6% positive) and fibroblasts (46.8% positive; Figure 3E and Appendix A). In the current experiments, neither FITC-GlaS nor APC-annexin A5 binding to PS-exposing m/lEVs derived from MDA-MB-231 cells was significantly different (56.4% or 50% positive, respectively) compared to non-cancerous cell-derived m/lEVs (Figure 3D,E and Appendix A). These results suggested that bead-based flow cytometric analysis using FITC-GlaS and APC-annexin A5 is limited to detecting PS-exposing m/lEVs released from certain cancer cell types. In addition, keratinocyte-derived sEVs and fibroblast-derived m/lEVs showed significantly increased binding of APC-annexin A5 and FITC-GlaS, respectively. Therefore, EV subtypes released from non-cancerous cells could be confounding for detecting and differentiating PS-exposing EVs released from cancer cells as a cancer biomarker.

### 3.4. Single EV Flow Cytometric Analysis of PS-Exposing sEVs (CD63^+^) and m/lEVs (Annexin A1^+^) Using FITC-GlaS or FITC-Annexin A5 Proteins

As noted above, the size of the sEV- and m/lEV-enriched fractions significantly overlap, and cross-contamination of EV subtypes is inevitable, posing a significant limitation of current EV isolation methods [34]. Therefore, we further characterized the distinct recognition of PS-exposing EV subtypes by FITC-GlaS or FITC-annexin A5 using a single EV flow cytometric analysis. Individual EVs were characterized using a highly sensitive spectral flow cytometer, the Cytek Aurora, with instrument settings optimized for detecting fluorescently labeled nanoparticles. Firstly, the isolated sEV- and m/lEV-enriched fractions from cancer and non-cancerous cells were stained with CellTrace Violet (CTV), an amine-reactive dye previously validated for nanoFACS [35]. Threshold triggering was performed using the fluorescence signature of CTV, which allowed for the detection of EVs and spectral unmixing of APC and FITC fluorescence signals. Next, the surface CD63, a commonly used marker for exosomes and/or sEVs [34], or annexin A1, a recently reported microvesicle and/or m/lEV marker [36], in individual sEVs or m/lEVs was detected using antibodies, respectively. The CTV-stained sEVs were incubated with APC-labeled anti-CD63 antibodies and either FITC-GlaS or FITC-annexin A5 proteins. In contrast, the CTV-stained m/lEVs were incubated with APC-labeled anti-annexin A1 antibodies and either FITC-GlaS or FITC-annexin A5 (Figure 4A). Samples were analyzed for CD63 and PS among sEVs and annexin A1 and PS among m/lEVs derived from the four cell types tested. There were no significant differences in either FITC-GlaS or FITC-annexin A5 detection of PS exposure on sEVs between cancer and non-cancerous cells under these experimental conditions (Figure 4B,C and Appendix A). In contrast, while not statistically significant, of all events acquired, the average percentages of annexin A1^+^ m/lEVs that also bound FITC-GlaS proteins were slightly higher in the MDA-MB-231 and MDA-MB-468 cells (23.8% and 28.3%, respectively), compared to keratinocytes (18.4%) and fibroblasts (21.3%; Figure 4D and Appendix A). Notably, the levels of FITC-annexin A5 binding were significantly higher on cancer cell-derived m/lEVs than on non-cancerous cell-derived m/lEVs. The percentages of acquired events that bound APC-anti-annexin A1 antibodies and FITC-annexin A5 were significantly higher in samples derived from MDA-MB-231 and MDA-MB-468 cells (33.3% and 43%, respectively) compared to keratinocytes (20.3%) and fibroblasts (18.5%; Figure 4E and Appendix A). These data suggest that m/lEVs could be more reliable EV subtypes for detecting PS exposure than sEVs, possibly due to their larger surface area. In addition, FITC-annexin A5 recognized more PS exposure on m/lEV subpopulations that contained annexin A1 compared to PS recognized by FITC-GlaS. However, the mechanism of their distinct recognition of PS-exposing EVs requires further study on other m/lEV subpopulations (e.g., BSG^+^ and SLC3A2^+^ [37]). Notably, we cannot rule out that two different PS-binding proteins affect PS localization on the purified EVs during the EV labeling, resulting in decreased stability of their PS binding on the EV surface.

## 4. Discussion

EVs have been actively studied as promising sources of cancer biomarkers for early diagnosis, prognosis, and monitoring therapy responses [15,38,39]. However, EV heterogeneity poses a significant challenge to developing reliable EV-based biomarkers; moreover, standardized EV isolation and profiling methods are still lacking. In addition, while new technologies for sensitive EV detection have been developed [40,41], they are difficult to implement widely in laboratories and hospitals since these systems often require specialized equipment and knowledge.

Flow cytometry is an alternative method commonly used for cellular analyses. Several EV characterization approaches using flow cytometry have been investigated. However, its potential for reliable EV analyses is still limited due to EVs’ sub-micron sizes, polydispersity, and low scattering properties relative to cells—EVs are more than 100 times smaller than cells [42]. A commonly used detection method is to capture isolated EVs with beads and analyze them using a conventional flow cytometer. However, these bulk EV analyses are suboptimal in addressing the heterogeneity of EVs to identify disease-associated EVs. To overcome this limitation, high-resolution flow cytometric methods to characterize individual EVs have emerged recently using combinations of efficient EV labeling strategies and high-resolution flow cytometry techniques [35,43]. Herein, we present a direct comparison between a bead-based EV assay and a single EV flow cytometry to assess PS-exposing EV subtypes, sEVs and m/lEVs, using two distinct PS-binding proteins, GlaS and annexin A5. The bead-based EV assay used in this study showed different binding patterns between sEVs and m/lEVs. This finding emphasizes the significance of EV subtype-specific characterization for developing reliable biomarkers. Notably, in our bead-based assays, both GlaS and annexin A5 proteins detected higher PS exposure on m/lEVs derived from MDA-MB-468 cells but not MDA-MB-231 cells (Figure 3D,E), suggesting the high PS exposure on EVs may be cancer cell type-dependent. Unexpectedly, higher PS display on sEVs derived from keratinocytes and m/lEVs from fibroblasts was detected using annexin A5 and GlaS, respectively (Figure 3C,D). This finding suggests that GlaS may recognize unique PS-exposing EVs relative to annexin A5. Here, we concluded that both GlaS and annexin A5 could recognize certain cancer cell-derived m/lEVs. However, interestingly, there could be different PS-positive EV populations, and these proteins might further elaborate different EVs within the sEV and m/lEV subtypes. Further investigation is needed to test this exciting possibility.

The single EV flow cytometric analysis under these experimental settings enabled the detection of higher PS exposure on m/lEVs derived from MDA-MB-231 cells and MDA-MB-468 cells relative to non-cancerous cell-derived m/lEVs. Interestingly, the bead-based EV assay did not show higher PS-exposing m/lEVs released from MDA-MB-231 cells. This finding emphasizes the significance of EV subtype-specific characterization to detect reliable biomarkers. Conversely, we detected no significant difference in PS display in sEVs between cancer and non-cancerous cells. However, we cannot rule out that single EV flow cytometric analysis lacks sensitivity and cannot resolve minor differences in PS exposure on sEVs due to their smaller surface areas compared to m/lEVs. In addition, since recent studies have shown that both sEV subpopulations originating from endosomes and plasma membranes contain CD63 [37], PS-exposure properties could differ depending on their original cellular compartments. Our data show CD81 and TSG101 could be more specific for sEVs (Figure 1D), so further studies on sEV subpopulations are needed to conclude. Notably, single EV flow cytometry is limited by the ability to resolve nano-sized biological particles using light scatter properties. While fluorescent labeling of EVs allows for the detection of nano-sized particles that fall under the light scatter detection limit, we cannot ensure that all EVs are evaluated in an individual sample. Other particles labeled sub-optimally may exist below the detection limit of the current instrument configuration. Further refinement of single EV analysis techniques will extend the capacity to characterize minor EV subtypes and non-EV extracellular particles [44], including exomeres and supermeres [45,46].

Accumulating evidence suggests that, unrelated to apoptosis, tumor cells can externalize PS on their surfaces [3,4,5,6]. Previous studies have shown that PS externalized on the surface of tumor cells contributed to the inhibition of T-cell-mediated tumor clearance [47,48]. Alternatively, masking PS on the cell surface prevented tumor growth [49]. In addition, PS is externalized on the surfaces of EVs released from viable tumor cells, and PS-exposing EVs can facilitate tumor growth, while the underlying mechanisms remain largely unknown [50]. Circulating EVs exposing PS in plasma were previously shown to be potential biomarkers for early cancer detection in clinical studies [17]. A recent in vivo imaging study of hepatocyte-derived sEVs demonstrated that macrophages eliminated the PS-exposing sEVs from the circulation [51] but not PS-deficient sEVs selected by the Tim4-based affinity selection [52,53]. Macrophages may help maintain the circulation of low PS-positive EVs in a healthy state. This finding is consistent with studies implicating elevated PS-exposing EVs as potential biomarkers for various human diseases [54,55], albeit further studies are needed to test this hypothesis.

## 5. Conclusions

Previous studies have demonstrated that tumor cells uniquely secrete PS-exposing EVs, which present as potential cancer biomarkers for early diagnosis. Here, we characterized the PS-displaying sEVs and m/lEVs recognized by the two PS-binding proteins, annexin A5 and GlaS, under the limitation of clear separation of EV subtypes. In the bead-based bulk EV assay, the MDA-MB-468 cell-derived m/lEV fraction showed significantly higher PS exposure recognized by both annexin A5 and GlaS proteins relative to the m/lEV fraction from non-cancerous cells. However, higher PS exposure on keratinocyte-derived sEVs and fibroblast-derived m/lEVs was also recognized by annexin A5 and GlaS proteins, respectively. In the single EV assay, we detected more PS-exposing m/lEVs from both cancer cell lines than non-cancerous cells by dual-labeling m/lEVs with anti-annexin A1 antibodies and PS-binding proteins. However, in our experimental setting, only annexin A5 showed consistently higher binding to cancer cell-derived m/lEVs (annexin A1^+^). GlaS showed the same tendency as annexin A5 and slightly higher signals in cancer cell-derived m/lEVs (annexin A1^+^) but was not statistically significant. Although non-immortalized, non-transformed primary cell types were compared to breast cancer cells, these controls were not derived from breast epithelial tissues. Therefore, we cannot rule out that an additional control, such as primary mammary epithelial cells, would have provided more consistent correlations relating to PS-exposing sEVs and m/lEVs. Although further extensive studies are needed to establish PS-exposing EV subtypes as reliable biomarkers for the early detection of cancer and other diseases, the single EV assay appears as a better tool to achieve this goal.

## Figures and Tables

**Figure 1 vaccines-11-00639-f001:**
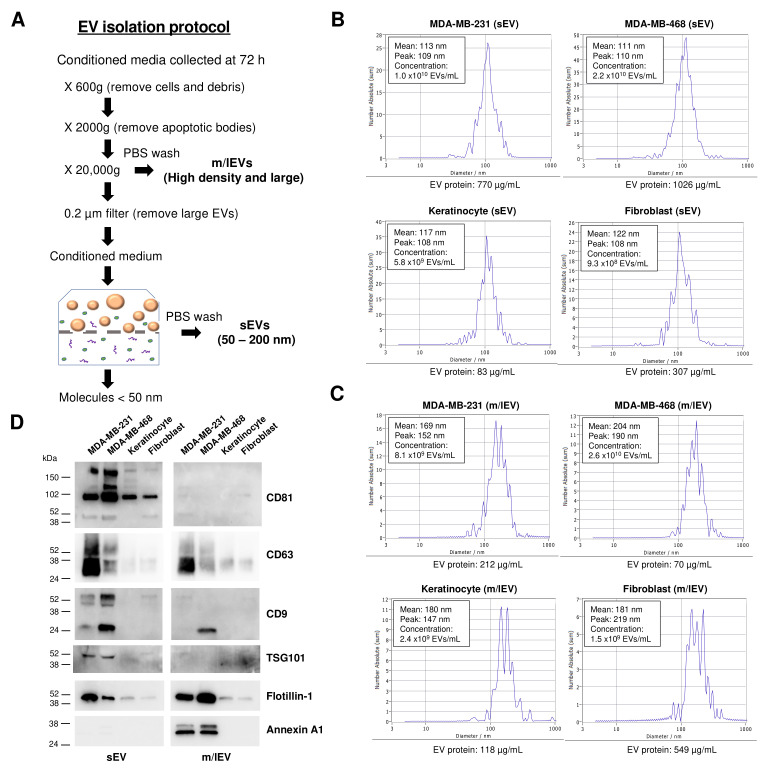
EV subtype-specific enrichment and characterization. (**A**) Schematic depiction of the isolation protocol for small EVs (sEVs) and medium/large EVs (m/lEVs). (**B**,**C**) Size distributions and protein concentrations of sEV- and m/lEV-enriched fractions derived from cancer (MDA-MB-231, MDA-MB-468) and non-cancerous cells (keratinocyte, fibroblast) measured by nanoparticle tracking analysis (NTA) and Bradford assay, respectively. (**D**) Western blot analysis of the EV marker proteins, CD81, CD63, CD9, TSG101, flotillin-1, and annexin A1 in sEV- and m/lEV-enriched fractions from each cell type.

**Figure 2 vaccines-11-00639-f002:**
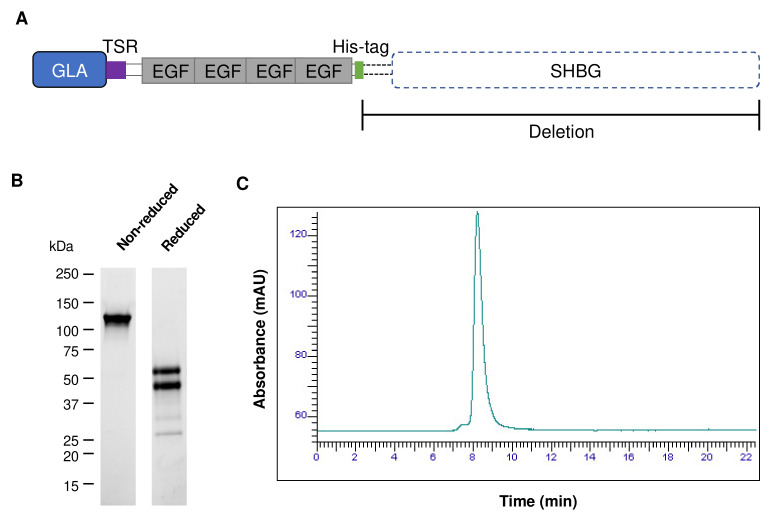
Design and production of the recombinant GlaS protein. (**A**) The GlaS protein was designed from a coagulation factor, Protein S, which consists of a vitamin K-dependent carboxylation/gamma-carboxyglutamic acid (GLA) domain, a thrombin-sensitive region (TSR), four epidermal growth factor (EGF)-like domains, and a sex hormone binding globulin (SHBG)-like domain. The C-terminal SHBG domain was deleted, and a His-tag was attached for protein purification and detection. (**B**,**C**) The recombinant GlaS protein was analyzed by Western blotting under non-reduced or reduced conditions (**B**) and HPLC-SEC (**C**).

**Figure 3 vaccines-11-00639-f003:**
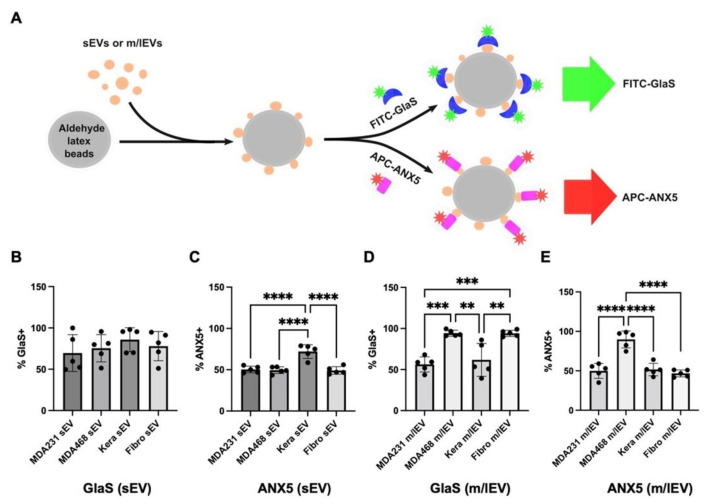
Bead-based analysis of phosphatidylserine (PS)-exposing sEVs and m/lEVs. (**A**) A schematic diagram of EV capturing on beads and analysis of their PS exposure by fluorescein isothiocyanate (FITC)-GlaS and allophycocyanin (APC)-annexin A5 (ANX5) proteins. (**B**–**E**) The beads capturing small EVs (sEVs) or medium/large EVs (m/lEVs) were labeled with FITC-GlaS or APC-ANX5 and analyzed by flow cytometry. The graphs show % of the FITC or APC positive EV-coated beads. Error bars, the standard deviation of five independent analyses, ** *p* < 0.01; *** *p* < 0.001; **** *p* < 0.0001.

**Figure 4 vaccines-11-00639-f004:**
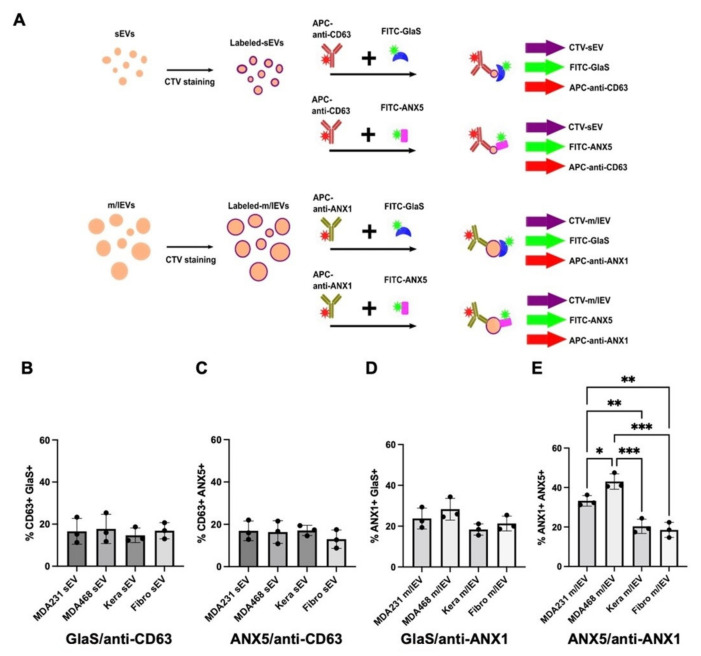
Single EV analysis of phosphatidylserine (PS)-exposing small EVs (sEVs) and medium/large EVs (m/lEVs). (**A**) A schematic diagram of EV labeling with CellTrace Violet (CTV), followed by incubating with antibodies against EV markers and PS-binding proteins. (**B**,**C**) CTV-sEVs were incubated with allophycocyanin (APC)-anti-CD63 antibodies and fluorescein isothiocyanate (FITC)-GlaS or FITC-annexin A5 (ANX5) proteins. (**D**,**E**) CTV-m/lEVs were incubated with APC-anti-annexin A1 (ANX1) antibodies and FITC-GlaS or FITC-ANX5 proteins. The graphs show % of the FITC/APC double-positive EV populations. Error bars, the standard deviation of three independent analyses, * *p* < 0.05; ** *p* < 0.01; *** *p* < 0.001.

## Data Availability

No new data were created.

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
