# Peer review of "Phosphatidylserine-Exposing Annexin A1-Positive Extracellular Vesicles: Potential Cancer Biomarkers"

_vaccines, 2023, doi:10.3390/vaccines11030639_

Round 1
Reviewer 1 Report (Previous Reviewer 2)
The authors addressed the previous critique adequately.
Reviewer 2 Report (Previous Reviewer 1)
The authors have addressed most of the issues that came up in the review
This manuscript is a resubmission of an earlier submission. The following is a list of the peer review reports and author responses from that submission.
Round 1
Reviewer 1 Report
In this article (Vaccines-2100583) entitled “Phosphatidylserine-exposing medium/large extracellular vesicles: Potential cancer biomarkers” Perez and colleagues report on a biochemical and cell biological study to isolate and characterize two types of extracellular vesicles (small extracellular vesicles (sEVs) and medium/large extracellular vesicles (m/lEVs)) from two breast cancer cell lines (MDA-MB-231 and MDA-MB-468) and two non-transformed cell lines (keratinocytes and fibroblasts). Authors first isolate above-mentioned EVs and then characterize associated EV biomarkers such as CD63, CD81, flotillin, others, and then perform a series of cell biological experiments to assess externalized PS and localization of two PS-targeting recombinant fusion proteins (FITC-Gla and APC AnxV). Main conclusions in the article include findings that (i) cancer cells secrete enhanced sFVs and m/lEVs compared to non-transformed cells, (ii) FITC-Gla and APC AnxV bind to different “speckles” on the m/lEVs, and (iii) EVs and be captured on aldehyde beads or with marker mAbs to potentially assess and quantify constitutive PS externalization.
Overall, this is a potentially interesting approach, although a bit descriptive and survey with low rigor. Authors conclude that different types of EVs can be determined from cancer cells by differential centrifugation, and that such EVs may have different levels of externalized PS. Cancer cells may also quantitatively secrete more EVs that non-cancer cells. While nothing objectionable in the paper, a few issues came up in thee review.
In Fig. 1, authors compare MDA-MB-231 and MDA-MB-468 versus keratinocytes and fibroblasts as the rationale for the study. Ideally, a parental non-transformed versus transformed line would be a better comparison. For example, do epithelial cell, in general, secrete more EVs than mesenchymal cells? A non-transformed epithelial cell such as MCF10A or a primary line would be a better control.
In Fig. 1, a loading control or some assessment of equivalent starting cells is needed to validate findings.
The data in Figure 3 is not so convincing. What are these speckles and can they be competed with PS liposomes?
The data in Fig. 4 and 5 is potentially interesting but a bit preliminary. What assurances are in place that PS localization is maintained in the sample preparation.
Reviewer 2 Report
In this manuscript, authors isolated sEVs and m/EVs from two breast cancer cell lines (MDA-MB-231 and MDA-MB-468) and non-cancerous cells (Keratinocytes and fibroblast) and compared the concentration and size among them. The author stated that tumor cell-derived extracellular vesicles (EVs) that exposed phosphatidylserine could be a potential biomarker for the early detection of cancer and other diseases. This study used recombinant proteins of annexin A5 and the carboxylated glutamic 26 acid domain of Protein S (GlaS), specific for PS, to detect PS-exposing EVs, and performed bead-based EV assays and single EV flow cytometry in enriched small EVs (sEVs) and medium/large EVs (m/lEVs) from breast cancer cells, keratinocytes, and fibroblasts to show that m/lEVs could be more reliable biomarkers compared to sEVs.
Although sEVs and m/EVs were used for comparison, there is significant variations in the overall size of sEVs and m/EVs. Further, the authors used single flow cytometry analysis to PS-exposing sEVs and m/IEVs using CD63 and annexin A1 markers respectively, and the authors did not find any difference in the percentage of sEVs between cancerous and non-cancerous cells; however, the authors found some statistically significant difference among cancerous and non-cancerous cells with ANX5 subtypes. Although the study on EVs may be important for the field of cancer, the comparison among different cell types is not scientifically justified and unfortunately, the majority of the results did not demonstrate statistically significant differences among cancerous and non-cancerous cell types. Accordingly, the conclusion drawn from this study was not clearly supported.
Comments:
Authors isolated sEVs and m/EVs from cancerous cells (MDA-MB-231 and MDA-MB-468) and non-cancerous cells (keratinocytes and fibroblast) and concluded that cancer cells produce a higher number of sEVs and m/IEVs relative to non-cancerous cells. Also, based on a comparison between cancerous and non-cancerous cells, authors concluded that cancerous cells induce more m/IEVs compared to non-cancerous cells which could be used as a biomarker. The whole manuscript is based on the results of a comparison between breast cancerous cells and non-cancerous cells (Keratinocytes and fibroblast). However, this comparison is not scientifically justified. It is not ideal to compare EVs produced from breast cancer cells with EVs produced from keratinocytes and/or fibroblasts. The authors, at least, should use cells derived from the same tissue as comparisons.
Similar to the above comment, the authors should also justify why the authors choose breast cancer cells and keratinocytes, and fibroblast instead of using the same cancerous and non-transformed cells in the text. The authors should also consider if some of the inconsistencies in results may be caused by the effect of various cells that has been used.
In Fig. 1D, authors did western blotting for the EV markers and concluded that sEVs derived from cancer cells show high enrichment of EV marker protein compared to non-transformed cell-derived EVs. However, the authors did not mention whether they used equal amount of protein or equal concentration of EVs.
In Fig. 4 and Fig. 5, the authors mentioned the number of biological replicates, however, the legends section in the figures does not show the number of experimental replicates. It seems that the authors have only used one experimental replicate which is not sufficient to make a conclusion. The Legends section of Fig. 5 states n=4, however in the figure, only 3 are visible.
In Fig. 4, the authors used CD63 as a marker for sEVs, however in Fig. 1D, they detected the CD63 signal in both sEVs and m/EVs. This creates a question over CD63 specificity for sEVs. Also, the authors used annexin A1 for m/EVs, however, there is no western blot to confirm the claim.
The p-value on the figures is missing.
This manuscript mentioned that fluorescently labeled GlaS and annexin A5 proteins recognize distinct EV subpopulations. However, the reference is based on unpublished work, thus it is difficult to evaluate the significance.
Minor comments:
1. Based on the result section described in lines 221-222, the study was not able to divide EVs based on size into small and large subtypes definitely. The authors may consider changing the title.
2. The line after 81 in the introduction, defining results may move to the result section.
3. The exact purpose of the manuscript in the introduction section is not very clear as to how this manuscript will help with challenges in establishing reliable biomarkers. The authors may expand the introduction to clarify.